# Influence of Applied Load and Sliding Velocity on Tribocorrosion Behavior of 7075-T6 Aluminum Alloy

**Zhengyi Li [1], Hongying Yu [2,3], Lei Wen [1,\*] and Dongbai Sun [1,3,4,\*]**

1   National Center for Materials Service Safety, University of Science and Technology Beijing, Beijing 100083, China
2   School of Materials, Sun Yat-Sen University, Guangzhou 510006, China
3   Innovation Group of Marine Engineering Materials and Corrosion Control, Southern Marine Science and Engineering Guangdong Laboratory (Zhuhai), Zhuhai 519082, China
4   School of Materials Science and Engineering, Sun Yat-Sen University, Guangzhou 510006, China
\*   Correspondence: wenlei@ustb.edu.cn (L.W.); sundongbai@mail.sysu.edu.cn (D.S.);
    Tel.: +18612707456 (L.W.); +0755-21096001 (D.S.)

**Abstract:** In this paper, the tribocorrosion behavior and synergistic effect of 7075-T6 aluminum alloy under different applied loads (100, 150, and 200 N) and sliding velocities (100, 150, and 200 rpm) were studied in a 3.5 wt.% NaCl solution. Tribocorrosion experiments were conducted in a tribobocorrosion system with pin-on-disc testing. The results show that the interaction between applied load and sliding velocity significantly affects the mechanical and electrochemical properties of 7075-T6 aluminum alloy. Increases in sliding velocity and applied load will accelerate the corrosion. Due to the synergistic effect of corrosion and wear, the wear rate is almost unchanged as the sliding velocity increases. When the applied load increased from 100 to 200 N, the wear rate increased from $1.97 \times 10^{-5}$ to $2.08 \times 10^{-5}$ mm$^3$/N·m, and the delamination wear phenomenon was aggravated.

**Keywords:** tribocorrosion; aluminum alloy; synergistic effect; electrochemical

## 1. Introduction

7075-T6 aluminum alloy has high impact toughness and broad application prospects in the marine equipment. However, aluminum alloys have a strong pitting tendency [1], especially among the corrosive chloride ions of the marine environment, which further accelerate the deterioration of aluminum alloys. Corrosion has been recognized as a hazard to aluminum alloys in the marine environment [2–4]. In some applications, such as marine drilling, corrosion and wear occur simultaneously, which accelerates the degradation of the material [5,6]. Tribocorrosion [7] can be considered as material degradation caused by the combined action of electrochemical corrosion and mechanical wear processes, such as sliding, rolling, impacting, fretting, corrosion, and the tribological process. Previous works have mostly focused on the corrosion resistance and microstructure of an aluminum alloy. However, there is rarely concern about the tribocorrosion of aluminum alloys [8,9]. Tribocorrosion has a greater impact on materials than corrosion. The tribocorrosion current density of 2024 aluminum alloy is 2–3 orders of magnitude higher than that of corrosion [10]. The material loss caused by tribocorrosion is not a simple superposition of corrosion and mechanical wear. As hydrostatic pressure increases, corrosion accelerates wear and becomes the dominant factor of degradation [11]. The passive film of an aluminum alloy will be destroyed or locally destroyed during the tribocorrosion process. A.C. Vieira studied the tribocorrosion behavior of aluminum alloy in a 0.05 M NaCl solution under different conditions [12]. Wear has little effect on accelerating corrosion, which accelerates the wear process of materials [13,14].

Tribocorrosion is associated with many factors, and Mischler lists the factors that influence the tribocorroison system but ignores the interaction of wear and corrosion [6].

The influences of applied load and sliding velocity on the tribocorrosion of aluminum alloy in the seawater were seldom considered in previous works. In this study, applied load and sliding velocity were considered as variables, and other parameters remained unchanged to investigate their influences on the tribocorrosion process.

## 2. Materials and Methods

### 2.1. Materials

Aluminum alloy 7075-T6 was used as the experimental material. Its composition is shown in Table 1. The specimens' diameter was 69.77 mm, and the thickness was 6.35 mm. The specimens were put in acetone under ultrasound treatment for 10 min. The working surfaces of the specimens were ground by sandpaper paper of 600, 1000, and 1500 grit, and cleaned with deionized water and ethanol. $Si_3N_4$ was selected as the counter body, whose roughness was 0.2 μm, and the contact surface between the counter body and the specimen was a sphere of 2 mm.

**Table 1.** The element composition of 7075-T6 aluminum alloy, wt.%.

| Al | Zn | Mg | Cu | Fe | Si |
|----|----|----|----|----|----|
| 89.69 | 5.35 | 2.65 | 1.62 | 1.19 | 0.11 |

### 2.2. Experiments

2.2.1. Tribocorrosion Experiments

The tribocorroison tests were conducted in a highly hydrostatic tribobocorroson system with a pin disc. The details were introduced in a previous study [11]. To avoid potential interference and galvanic corrosion of contact between different metals, the specimen needs to be sealed. During each tribocorrosion test, a rotating $Si_3N_4$ counter body slides against the specimen in a 3.5 wt.% NaCl solution (simulated seawater). Aluminum alloy is very sensitive to the oxygen content, pH, and temperature of the solution. Therefore, the solution temperature, pH, and solution of dissolved oxygen (DO) content were controlled: DO of 6.68 ppm, pH of 6.47, solution temperature of 18 °C. The applied load and sliding velocity were used as variable conditions in the experiment. The sliding velocity was varied from 100 to 200 rpm (corresponding to frequency varying between 0.67 and 1.33 Hz), and the applied load was varied from 100 to 200 N. In the marine environment, the hydrostatic pressure is bound to bring an unexpected but inevitable environmental load. Therefore, an external load was added to offset the hydrostatic pressure in this experiment to ensure the accuracy of the contact load. To monitor the potential and current changes of 7075-T6 aluminum alloy, the experimental device was connected to an electrochemical workstation. In the electrochemistry system, the sample was used as a working electrode. The counter electrodes and reference electrodes were platinum, and Ag and AgCl (saturated KCl solution), respectively. The detailed electrochemical information was obtained through an electrochemical workstation (Princeton P3000, AMETEK, Berwyn, PA, USA) to monitor the corrosion behavior in situ. All the electrochemical experiments were performed in a 3.5 wt.% NaCl solution.

2.2.2. Electrochemical Experiments

The tribocorrosion test and pure mechanical wear test were carried out by controlling the potential. The tribocorrosion tests were carried out under open circuit potential (OCP). The pure mechanical wear test can inhibit the corrosion of aluminum alloy by applying the cathodic protection potential to achieve the state of no corrosion and obtain the pure mechanical wear amount. The self-corrosion potential ($E_{corr}$) of aluminum alloy was first determined by measuring the potentiodynamic polarization curve of the tribocorrosion experiment. In addition, as pure mechanical wear and tribocorrosion experiments, the OCP of the material surface and the polarization curve of the tribocorrosion process were simultaneously measured, and the electrochemical information, such as the self-corrosion

current density and self-corrosion potential of the material under the condition of no wear and the condition of wear, were obtained. To ensure a high accuracy of the electrochemical information, the sample was first immersed for 30 min before each test to keep the OCP stable. The whole friction process lasted for 10 min. After rubbing, the test was continued for 10 min to observe the OCP information. The polarization curve was tested only during the rubbing, the scanning rate of potential was 1 mV/s, and the potential scanning range was −500~250 mV vs. OCP.

### 2.2.3. Microstructure

The morphology of the specimen was observed by field emission scanning electron microscope (FESEM, Merlin Compact, ZEISS, Oberkochen, Germany). The surface components of the tested samples were characterized by X-ray energy spectrum (EDS, Oxford Instruments, Abingdon, UK). The total volume of the wear track was measured by 3D measuring laser microscopy (OLS4100, OLYMPUS, Tokyo, Japan).

## 3. Results and Discussion

After the tribocorrosion experiment, the surface of the sample was washed with alcohol and deionized water, and the sample was dried and vacuum treated for backup use. The real picture after tribocorrosion is shown in Figure 1. The morphology of the wear track is one of the important pieces of evidence for the tribocorrosion mechanism of materials. Figure 2 shows the obtained 2D and 3D information on the wear track contour of specimens by 3D measuring laser microscopy.

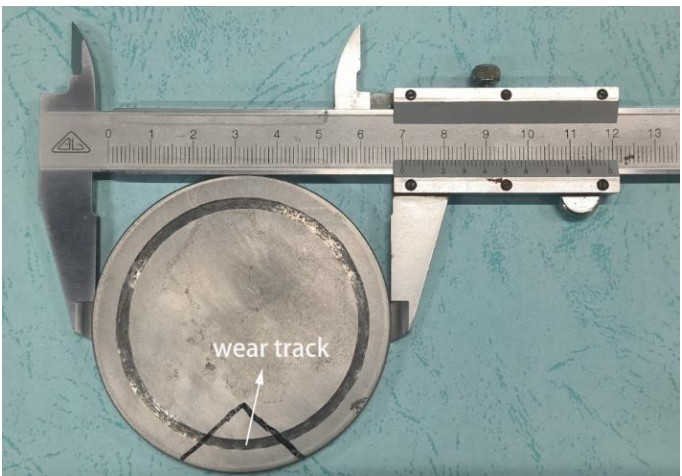

**Figure 1.** 7075-T6 aluminum alloy after tribocorrosion test.

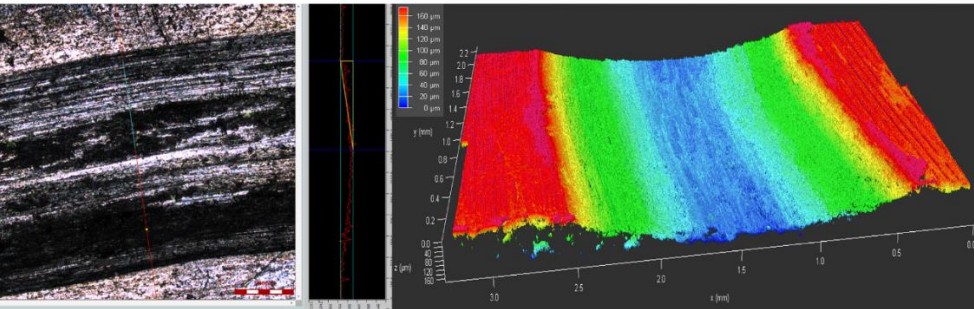

**Figure 2.** The 2D and 3D information of the wear track contour.

*3.1. Tribocorrosion Test*

3.1.1. Coefficient of Friction

The coefficient of friction reflects the friction force ratio between the wear contact surfaces and the vertical force acting on them. This is the result of the comprehensive action of many factors, such as the performance of the counter body, the geometric characteristics of the contact surface, and the magnitude of the applied load. The instantaneous coefficient of friction under three different sliding velocities is shown in Figure 3 and Table 2. It can be seen that the coefficient of friction greatly fluctuates at a high sliding velocity, compared with the low sliding velocity of 100 rpm. The average coefficient of friction increased from 0.41 to 0.55 when the sliding velocity increased from 100 to 200 rpm. The variation of the instantaneous coefficient of friction under different applied loads is shown in Figure 4. It can be seen that the average coefficient of friction increases when the applied load increases from 100 to 150 N. The coefficient of friction decreases by 16.36% when the applied load varies between 100 and 200 N. It can also be observed that the fluctuation of the coefficient of friction is not obvious under a high applied load, compared with the low applied load of 100 N.

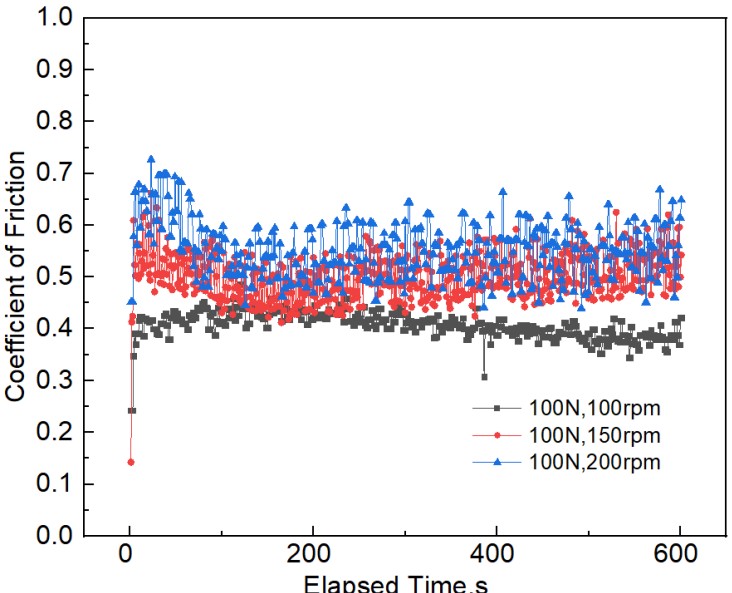

**Figure 3.** The instantaneous coefficient of friction between 7075-T6 aluminum alloy and $Si_3N_4$ under different sliding velocities.

**Table 2.** Average coefficient of friction in Figures 3 and 4.

| Parameter | 200 rpm | | | 100 N | | |
| --- | --- | --- | --- | --- | --- | --- |
| | 100 N | 150 N | 200 N | 100 rpm | 150 rpm | 200 rpm |
| COF | 0.55 | 0.45 | 0.46 | 0.41 | 0.50 | 0.55 |

3.1.2. Wear Rate

The tribocorrosion test of 7075-T6 aluminum alloy under different applied loads and sliding velocities was carried out. Figures 5 and 6 show the variation in wear rate with the variations in applied load and sliding velocity, respectively. The wear rate increases with the increase in applied load, as shown in Figure 5. When the applied load increases from 100 to 200 N, the wear rate increases from $1.97 \times 10^{-5}$ to $2.08 \times 10^{-5}$ mm$^3$/N·m. However, at different sliding velocities in seawater, no noticeable difference can be observed in Figure 6.

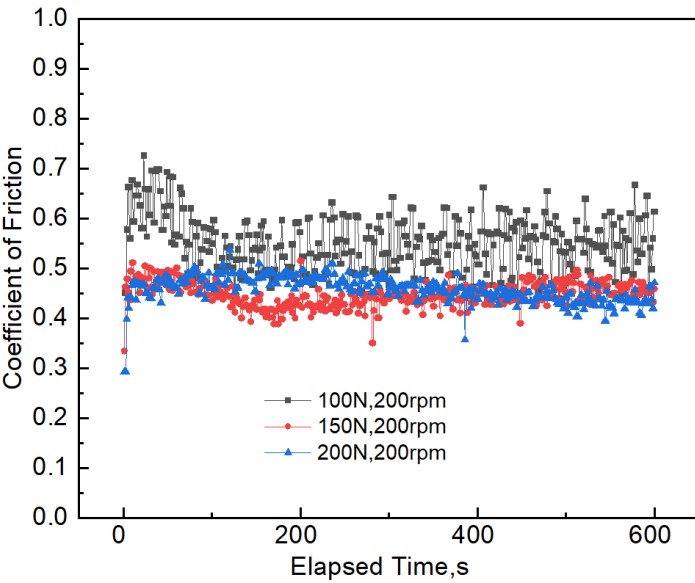

**Figure 4.** The instantaneous coefficient of friction between 7075-T6 aluminum alloy and Si$_3$N$_4$ under different applied loads and a constant speed of 200 rpm.

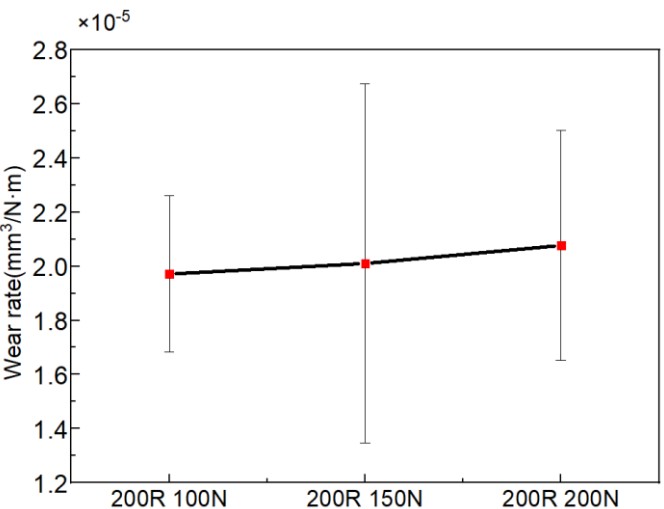

**Figure 5.** Variation in wear rate with the variation in applied load at a constant speed of 200 rpm.

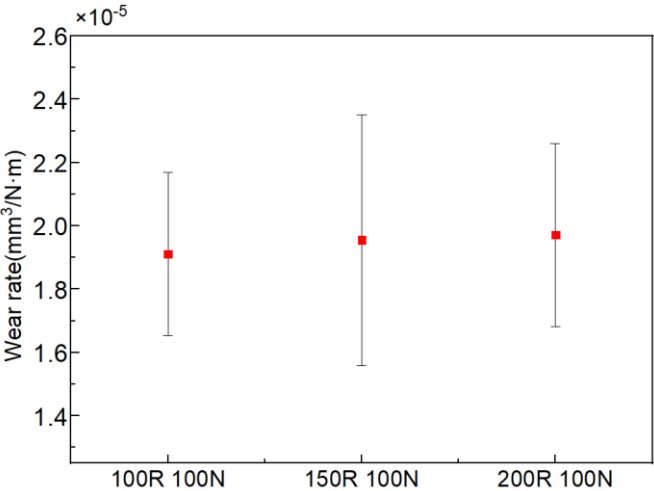

**Figure 6.** Variation in wear rate with the variation in sliding velocity at a constant load of 100 N.

### 3.2. Electrochemical Test

#### 3.2.1. Open Circuit Potential

Electrochemical corrosion will continuously occur when samples are immersed in seawater during the tribocorrosion. Therefore, it is very important to obtain the electrochemical corrosion information for a better understanding of tribocorrosion. The OCP reflects the electrochemical properties of the material surface. The OCP information under different applied loads is shown in Figure 7. When it reaches 600 s and the sliding starts, the OCP instantaneously decreases (negative potential shift) under the three sliding velocities. The larger the applied load, the more negative the OCP moves. When the load is 100, 150, or 200 N, the sliding process first decreases to the most negative point, then gradually increases to a steady state. When the rubbing stops, the OCP rapidly increases (positive potential shift) to the level close to that before the friction. Negative potential shift and positive potential shift are also mentioned in other studies on passive metals and alloys [15,16]. Due to the change in material surface state caused by friction, and due to the fact that the passive film needs a long time for passivation, the OCP after rubbing is lower than that before rubbing. Figure 8 shows the relationship between OCP and sliding velocity during tribocorrosion process in seawater. The trend is consistent with the change observed in Figure 7. In addition, the larger the sliding velocity, the more positive the OCP.

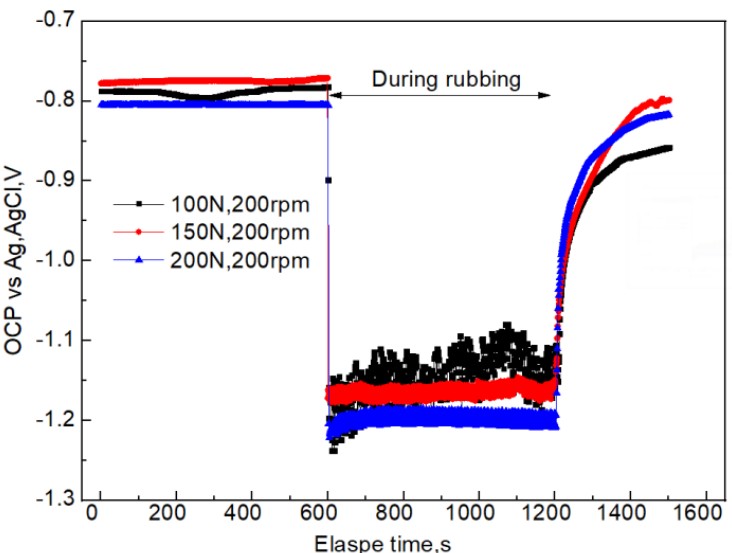

**Figure 7.** OCP information before, during, and after friction of 7075-T6 aluminum alloy and a silicon nitride counter body under different applied loads and a constant speed of 200 rpm.

#### 3.2.2. Potentiodynamic Polarization Curve

The corrosion dynamic information can be obtained from the polarization curve. The polarization curves of the tribocorrosion under different applied loads and sliding velocities are shown in Figures 9 and 10, respectively. The corresponding self-corrosion current density and self-corrosion potential information of polarization curves are presented in Table 3. By analyzing the data in Table 2, it can be deduced that the self-corrosion current density of aluminum alloy 7075-T6 increases with the increase in the applied load (Figure 9). When the applied load increases from 100 N to 200 N, the self-corrosion current density increases from 172.37 to 197.48 $\mu A/cm^2$, which denotes an increase of 14.57%. Under the constant applied load of 100 N, when the sliding velocity increases, all these curves show a similar trend to Figure 9. However, the self-corrosion current density of aluminum alloy slightly increases in Figure 10.

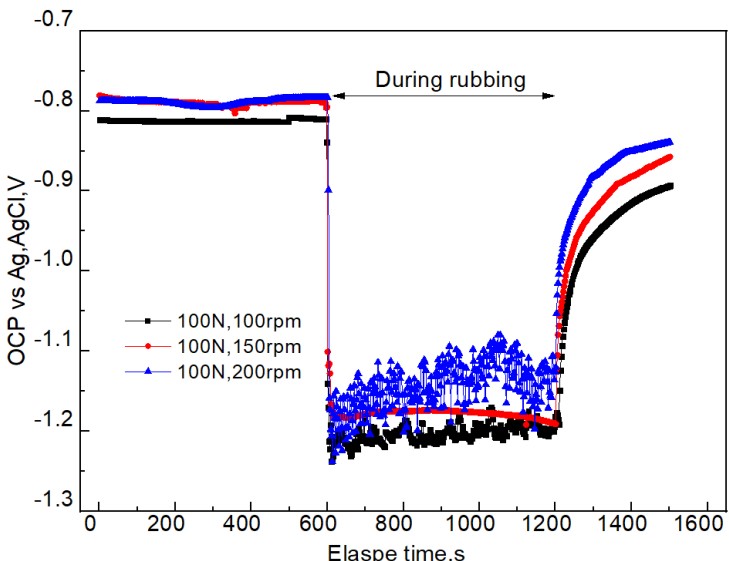

**Figure 8.** OCP information before, during, and after friction of 7075-T6 aluminum alloy and a silicon nitride counter body under different sliding velocities and a constant load of 100 N.

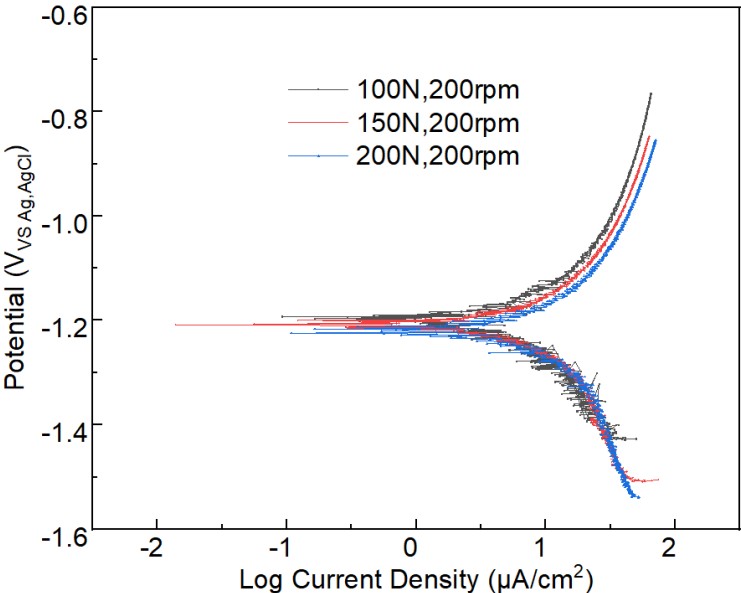

**Figure 9.** Polarization curve information of the tribocorrosion process under different applied loads and a constant speed of 200 rpm.

**Table 3.** The electrochemical information is shown in Figures 10 and 11.

| Parameter | Current Density, $\mu A/cm^2$ | Potential, V $_{VS. Ag, AgCl}$ |
|---|---|---|
| 100 rpm 100 N | 169.81 | $-1.158$ |
| 150 rpm 100 N | 170.92 | $-1.178$ |
| 200 rpm 100 N | 172.37 | $-1.167$ |
| 200 rpm 150 N | 183.09 | $-1.201$ |
| 200 rpm 200 N | 197.48 | $-1.216$ |

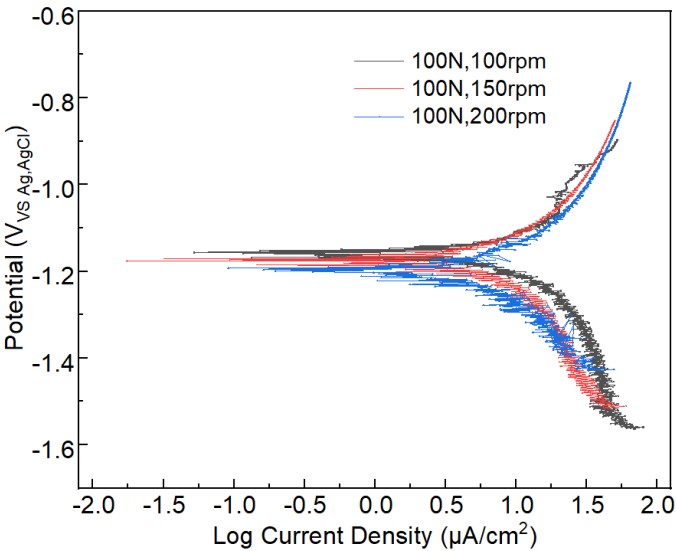

**Figure 10.** Polarization curve information of the tribocorrosion process under different sliding velocities and a constant load of 100 N.

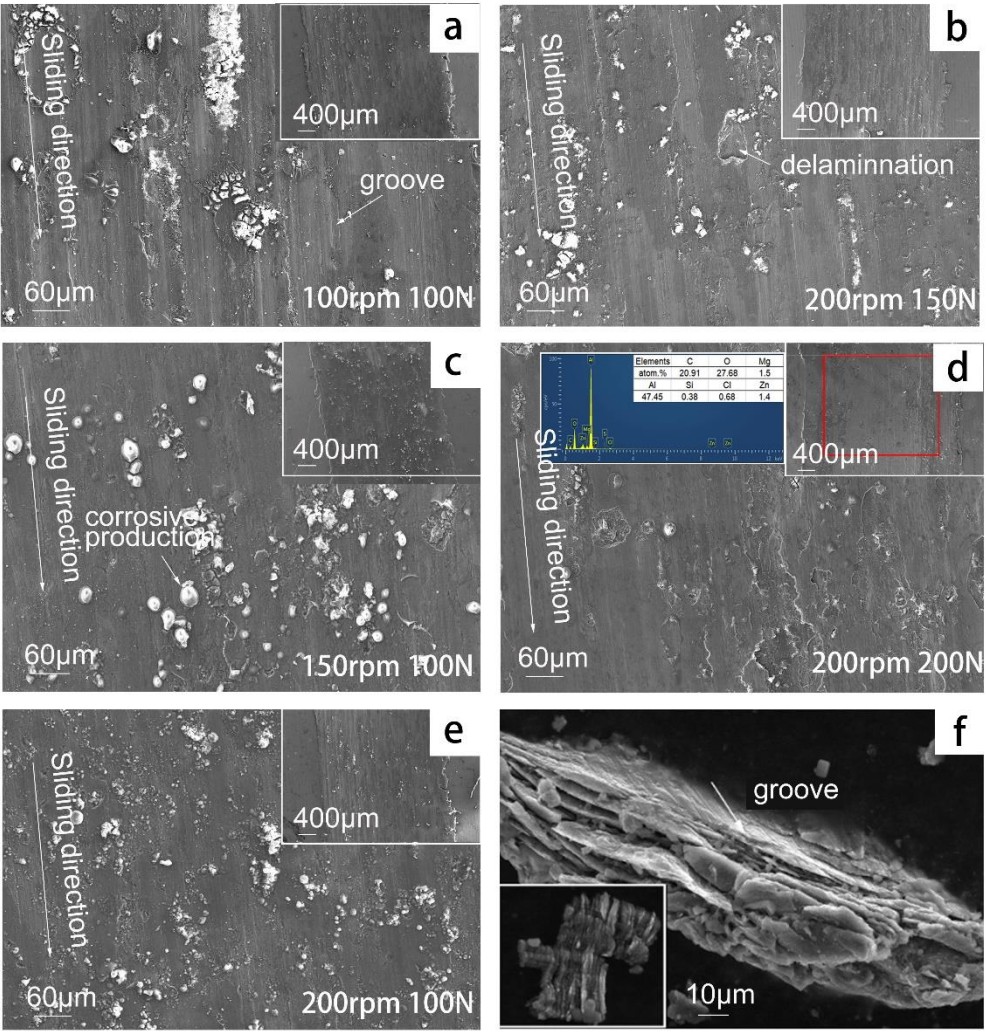

**Figure 11.** Tribocorrosion microstructure of 7075-T6 aluminum alloy under different wear conditions: (**a**) 100 rpm, 100 N; (**b**), 200 rpm, 150 N; (**c**) 150 rpm, 100 N; (**d**) 200 rpm, 200 N; (**e**) 200 rpm, 100 N; (**f**), debris.

### 3.3. Microstructure of the Wear Track

The microstructure of a wear track is important evidence for understanding the tribocorrosion mechanism. Figure 11 presents the micro-morphology of the worn area under different applied loads (e,b,d) and different sliding velocities (a,c,e). The microstructure of the wear track shows obvious grooves, delamination, tearing edge, and plastic deformation. At the edge of the wear track, the material has been plowed outside the wear track, which means that there was serious plastic deformation during tribocorrosion. It can be deduced from the EDS of the wear track that the distribution of oxygen element in the wear track is obvious, and the distribution of chlorine element is also in the wear track, which indicates that corrosive chloride ions and oxygen play important roles during the tribocorrosion. After the sliding velocity was increased, both grooves and delamination were visible in the worn area, and more corrosion products were distributed inside the worn area. When the applied load was increased, less grooves appeared, but more serious delamination phenomena were visible, and no uniform corrosion on the whole surface occurred. After a larger load of 150 or 200 N was applied to the aluminum alloy, there were cracks around the delamination and the groove. The micro-morphology of the wear track shows the coexistence of various wear mechanisms, and the wear mechanism of aluminum alloy did not change because of the sliding velocity and applied load, which still showed fatigue wear and delamination wear with corrosion. However, with the increase in the applied load, the delamination wear became more aggravated.

## 4. Discussion

Tribocorrosion is a process in which corrosion and wear coexist. Corrosion, wear, and their synergistic effects should be carefully analyzed to better understand tribocorroison. The corrosion of the aluminum alloy cannot be ignored when it is immersed in seawater. Chlorine and oxygen elements are adsorbed on the wear track surface of aluminum alloy, indicating that the surface of aluminum alloy has been corroded, as shown in Figure 2.

In the initial phase of friction, the instantaneous friction coefficient has its lowest value. There are several reasons for that phenomenon: (1) The aluminum alloy formed a stable passivation film in the air, which has a certain lubrication effect. As the oxide film easily separates the surfaces of the two materials, with little or no true metal contact, the oxide film has a low shear strength. (2) Seawater is not only a corrosive solution; it has a lubricating effect, which reduces the coefficient of friction between materials [17]. (3) The corrosive products on the surface of the aluminum alloy are considered to be responsible for the friction reduction (Figure 3). In the tribocorrosion process, the friction coefficient gradually increased and reached a stable state at about 120 s. The reason for the increase in the coefficient of friction was that the roughness increased and a large amount of wear debris occurred [18]. The coefficient of friction of the aluminum alloy increased with the increase in the sliding velocity, which may have been due to the fact that the change in shear rate affected the mechanical properties of the material. Previous studies have shown greater strength of materials at higher shear strain rates [19], which results in lower actual contact areas and lower friction coefficients under dry contact conditions. However, the trend is indeed reversed in corrosive environments. This indicates that the corrosion alters the surface strength of the aluminum alloy and deteriorates its properties. The reason for the friction coefficient's increase with the sliding velocity may be that a higher sliding velocity leads to a higher surface corrosion rate [9], which further deteriorates the material's properties. The higher current density of the polarization curve under higher sliding velocities of 150 and 200 rpm also indicates that the corrosion rate is faster than 100 rpm. After the corrosion of the materials, the hardness is reduced [20], which results in a larger contact area between the aluminum alloy and the counter body. The coefficient of friction decreases with as the applied load increases. This is due to the fact that when the applied load increases, the local stress will rapidly increase, allowing the surface material to easily result in yield and fatigue, and finally fall off into debris under the action of normal stress, while the debris mainly acts as the role of "ball".

When the applied load increases, the normal force on the aluminum alloy increases, and the contact area of the counter body pressed into the aluminum alloy increases. Different degrees of plastic deformation occur in the material (Figure 11), leading to dislocation energy accumulation, increasing the possibility of chemical reactions and increasing the hardness [21]. The occurrence of corrosion will form a corrosion product on the surface of the material, and reduce or inhibit the deformation strengthening of the aluminum alloy while its shear strength decreases, which further leads to a higher wear rate (Figure 5). The increase in the velocity is equivalent to the increase in the friction at the same time, which is bound to lead to the increase in the wear rate [22], and it follows the same trend shown in Figure 6. It was also shown that the wear intensifies with the increase in the applied load and sliding velocity (Figure 11). Based on the above evidence, it can be inferred that the corrosion accelerates the wear of aluminum alloys.

After the initial friction, the film is damaged, and the fresh metal is exposed in the solution, which results in instantaneously reducing the OCP (Figures 7 and 8). This indicates that the corrosion tendency of the aluminum alloy is increased. The reason is that the dissolved oxygen and chloride ions in the solution can directly react with the fresh aluminum alloy, which accelerates the anodic dissolution of aluminum alloy, increases its self-corrosion current density, and results in negative potential movement [23]. When applying a larger load, the plastic deformation of samples was more serious, resulting in significant damage to the passive film and the OCP moving in the negative direction. After a period of friction, the roughness and other parameters may reach certain steady values, resulting in stable coefficient of friction and OCP. The self-corrosion current density had the same change trend (Figures 9 and 10). A possible reason is that the counter body, aluminum alloy, debris, and solution reached a relatively stable friction state. It is important to mention that there was a correlation between the OCP and the coefficient of friction during the tribocorrosion process. The OCP sharply fluctuated in the early stage. The coefficient of friction also greatly fluctuated, which also indicates that there was a synergistic effect between corrosion and wear (Figure 12).

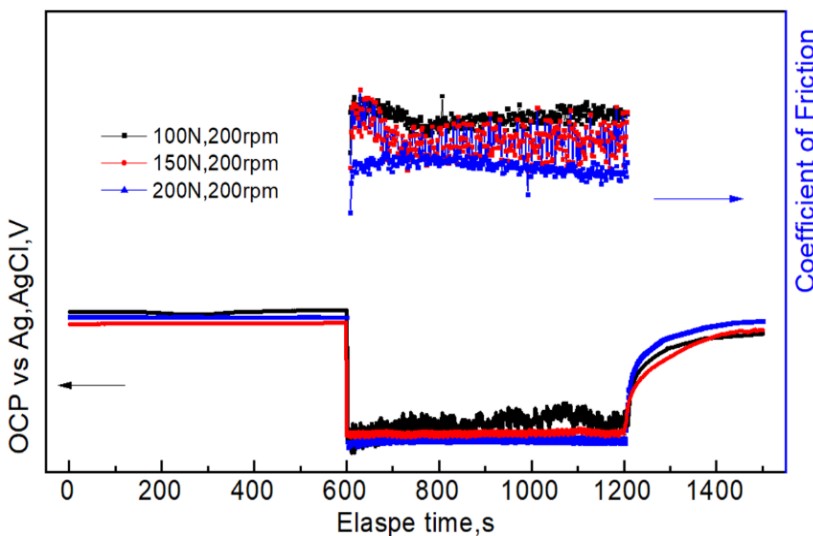

**Figure 12.** Correlation between coefficient of friction and OCP.

Applying a larger load is equivalent to increasing the contact area between the counter body and the aluminum alloy. It indicates that a larger area of bare aluminum alloy (the passive film is removed) is exposed. When the passive film is locally destroyed in the wear process, the corrosive ions in seawater can erode into the defects and change the friction behaviors of the material [24]. Therefore, more chloride ions and oxygen ions in the solution will participate in the anodic dissolution reaction of bare aluminum alloy, which is directly reflected in the increase in the self-corrosion current density. This is consistent

with the curve trend shown in Figure 10. At high loads, cracks exist around the groove and delamination (Figure 11). A possible reason is that microscopic crevice corrosion will occur because of electrochemical action, and it can develop into cracks under the action of alternating stress [10,21].

Faster sliding velocity is essentially for a shorter time interval between activation and passivation of the same location. This means that at the same time, the exposed bare metal area is larger, and the result is consistent with the effect of increasing the load. It will result in strengthening the anodic dissolution reaction of aluminum alloy and increasing the self-corrosion current density (Figure 10). It can also be seen in Figure 11 that the corrosion intensifies with the increase in applied load and sliding velocity. During the tribocorrosion, the passive film or metal is damaged or removed, and the exposed metal will quickly react to form a new passive film. Therefore, the tribocorrosion is a process involving repassivation and depassivation, which can be confirmed by the fluctuation of the polarization curve (Figure 10). Previous studies have deduced that the self-corrosion current density under tribocorrosion is twice as high as that under the absent of friction [11]. Combined with the above analysis, it is easy to understand the reason why wear accelerates corrosion.

### 5. Conclusions

(1) The interaction of applied load and sliding velocity significantly affects the mechanical properties of aluminum alloy sliding against $Si_3N_4$. When the load increases from 100 to 200 N, the wear rate increases from $1.97 \times 10^{-5}$ to $2.08 \times 10^{-5}$ mm$^3$/N·m. Due to the synergistic effect of corrosion and wear, the wear rate is unchanged with the increase in sliding velocity.

(2) The interaction of applied load and sliding velocity significantly accelerates corrosion. The self-corrosion current density increases as the applied load increases. When the load increases from 100 to 200 N, the self-corrosion current density increases from 172.37 to 197.48 $\mu A/cm^2$, which is 14.57%. The increase in sliding velocity affects corrosion less.

(3) The wear mechanism of the aluminum alloy does not change because of sliding velocity and applied load, which still result in fatigue wear and delamination wear with corrosion. However, with an increase in applied load, the delamination phenomenon becomes more serious.

**Author Contributions:** All the co-authors made a contribution to this paper. Z.L., the first author, was the designer and executor of the research and the writer of the article. H.Y., the second author, who designed the experiment and discussed the data. The third and the fourth author are corresponding authors. D.S. and L.W. who designed the experiment and discussed the results together. D.S. and L.W. are the project leaders who financially supporting this research. All authors have read and agreed to the published version of the manuscript.

**Funding:** This work was supported by National Key Research and Development Program of China (grant number 2017YFA0403000, 2017YFA0403404, and 2017YFA0403002), the National Natural Science Foundation of China (grant number U1837602, U21B2053 and 42076212), and Innovation Group Project of Southern Marine Science and Engineering Guangdong Laboratory (Zhuhai) (grant number 311021013).

**Institutional Review Board Statement:** Not applicable.

**Informed Consent Statement:** Not applicable.

**Data Availability Statement:** The raw/processed data required to reproduce these findings cannot be shared at this time as the data also forms part of an ongoing study.

**Acknowledgments:** The authors are grateful to the State Key Laboratory of Solid Lubrication, Lanzhou Institute of Chemical Physics, and Chinese Academy of Sciences for providing the research facilities.

**Conflicts of Interest:** The funders had no role in the design of the study; in the collection, analyses, or interpretation of data; in the writing of the manuscript; or in the decision to publish the results.

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
