# Peer review of "Influence of Applied Load and Sliding Velocity on Tribocorrosion Behavior of 7075-T6 Aluminum Alloy"

_metals, doi:10.3390/met12101626_

Round 1

Reviewer 1 Report

Dear Authors,

I would like to congratulate you for your work presented in manuscript titled "Influence of applied load and sliding velocity on tribocorrosion behavior of 7075-T6 aluminum alloy".

I've found the manuscript well written with good flow. The methodology of the study and the structure of the manuscript are clear. The results and discussion sections are well written and informative. 

My questions are as follows:

- I am not sure if and how Fig. 2 would add much to your manuscript. The EDS results are very noisy in general. The quality of image would not be good in print version nor in digital format at 100% scale. 

- The error bars in Fig. 6 and 7 are relatively large, especially the middle ones. How many samples were used for these two plots? The middle error bar is large enough to question the positive trend of the wear rate with load and sliding velocity.

- Please check the the manuscript thoroughly again to ensure it meets journal's requirements and implement any  grammatical modifications needed.

Sincerely

Author Response

Dear reviewer,

Thank you for taking time out of your busy schedule to review my manuscript. I have carefully read your comments and suggestions, and made modifications and replies one by one. The detailed changes are in the attachment.

Reviewer 2 Report

Row 40:  conditions, Wear (need to be corrected)

Row 54: sandpaper is used for grinding, not polishing. Polishing was not carried out?

Table 1: the elemental or element composition (I suppose).

Row 63-64: Sentence: To avoid - is unfinished

Row 67 - 68: How was contolled dissolved oxygen (DO)?

Row 72-73: Sentence: Therefore, an external ... is undestandable

Row 92: Sample was immersed - state where!

Row 100: And at the beginning of the sentence?

Fig. 2 shows morphology of wear track or topographical microstructure of wear track and elemental mapping. It need to be add under Figure - elemental mapping or EDS mapping...

Findings on rows 107-111: chlorine distirbution is not visible,. 

Row  134 -135:  Statement does not agree with the figure 5. The average coeffcient of friction increases with the increase of the applied load?

Row 153: applied and load?

How was it determined wear rate?

Fig. 7 is not included in the paper text.

I recommend  in the description of the figures (below the figure) add constant condition of experiment, for example at sliding velocity 200rpm

Author Response

(The authors gave the same response as above.)
